# LoRTA: Low Rank Tensor Adaptation of Large Language Models

## Abstract

Low Rank Adaptation (LoRA) is a popular Parameter Efficient Fine Tuning (PEFT) method that effectively adapts large pre-trained models for downstream tasks. LoRA parameterizes model updates using low-rank matrices at each layer, significantly reducing the number of trainable parameters and, consequently, resource requirements during fine-tuning. However, the lower bound on the number of trainable parameters remains high due to the use of the low-rank matrix model. In this paper, we address this limitation by proposing a novel approach that employs a low rank tensor parameterization for model updates. The proposed low rank tensor model can significantly reduce the number of trainable parameters, while also allowing for finer-grained control over adapter size. Our experiments on Natural Language Understanding, Instruction Tuning, Preference Optimization and Protein Folding benchmarks demonstrate that our method is both efficient and effective for fine-tuning large language models, achieving a reduction in the number of parameters while maintaining comparable performance.

## 1 Introduction

The advent of Large Language Models (LLMs) – billion parameter scale models pre-trained on vast corpora of data – has enabled unprecedented capabilities across a wide range of tasks. However, as LLM sizes continue to grow exponentially, their computational and memory demands represent significant challenges, particularly for those lacking access to high-performance computing infrastructure (Varoquaux et al., 2024). This has spurred interest in parameter efficient fine tuning (PEFT) techniques (Ding et al., 2023), which facilitate the adaptation of LLMs to specific applications, downstream tasks or user preferences, by using only a small fraction of trainable parameters. Most importantly, they reduce GPU memory requirements, primarily by shrinking optimizer states (Liao et al., 2023). Moreover, they provide greater efficiency in storage and deployment, enabling the management of multiple fine-tuned LLMs with reduced storage footprints and faster load times (Sheng et al., 2023; Wen & Chaudhuri, 2024), which is particularly relevant for applications requiring rapid model switching across numerous task- or user-specific models.

Beyond computational benefits, PEFT techniques can also mitigate overfitting risks associated with fine-tuning high-capacity LLMs. By constraining model updates, PEFT methods can act as an implicit regularization mechanism, improving generalization (Fu et al., 2023; Sun et al., 2023). Parameter sharing, a well-established technique in deep learning architecture design, has been shown to improve generalization across various tasks such as protein folding (Jumper et al., 2021; Lin et al., 2023), image segmentation (Ronneberger et al., 2015), and generative modeling (Rombach et al., 2022). Incorporating parameter sharing in PEFT methods has also improved performance in specialized applications with limited data, such as in medical domains (Dutt et al., 2023; Zhu et al., 2024).

Low Rank Adaptation (LoRA) is a popular PEFT approach that uses a low rank parameterization of weight matrix updates (Hu et al., 2021). For instance, these allow to fine tune a 175 billion parameter LLM using only 5 million trainable parameters (Hu et al., 2021) without performance degradation. Since the model updates can be merged with the frozen weights, LoRA incurs no additional inference cost when deployed, unlike prompt (Li & Liang, 2021a; Liu et al., 2023) and adapter-based (Houlsby et al., 2019; He et al., 2021; Pfeiffer et al., 2020) PEFT methods. However, the lower bound on trainable parameters often remains substantial for large-scale models. Recent works have aimed to further reduce the number of parameters in LoRA by allocating different ranks

across update matrices in different layers (Valipour et al., 2022; Zhang et al., 2023b), using fixed low rank projections (Zhang et al., 2023a; Kopiczko et al., 2023), and parameterizing low rank matrices using a random basis (Koohpayegani et al., 2024).

In this work, we introduce Low Rank Tensor Adapters (LoRTA), which exploit redundancy in weight updates across different layers, heads, and attention matrices by representing updates as a unified 5th-order low-rank tensor model. This holistic approach not only reduces the number of trainable parameters but also facilitates learning by exploiting shared information among various model components. We show that the parameter-sharing schemes in state-of-the-art methods (Kopiczko et al., 2023; Koohpayegani et al., 2024) are implicit low-rank tensor models with fixed random factors, while our explicit tensor model is *fully trainable* and more expressive. Our higher-order tensor update and Candecomp-Parafac model (Harshman & Lundy, 1994) enjoys greater parameter efficiency and favorable scaling over low-rank tensor models recently proposed for vision transformers (Jie & Deng, 2023; Edalati et al., 2023) and LLMs (Yang et al., 2024; Bershatsky et al., 2024).

The two main advantages of our proposed method over existing matrix and tensor based approaches can be summarized as follows:

(A1) It enables a reduction in the number of trainable parameters by using updates with lower tensor rank.

(A2) The number of parameters scales better with tensor rank, number of fine-tuned matrices, embedding dimension and attention heads.

As a consequence, our approach also provides finer-grained control of adapter size. We evaluate our method on diverse benchmarks including Natural Language Understanding, Instruction Tuning, Preference Optimization, and Protein Folding. Our experiments demonstrate that LoRTA can achieve up to an order of magnitude reduction in the number of trainable parameters compared to state-of-the-art PEFT methods, with minimal performance trade-offs.

## 2 PRELIMINARIES

### 2.1 TRANSFORMER ARCHITECTURE

We focus on the transformer architecture, although it can be naturally extended to other architectures such as Convolutional Neural Networks and Long Short Term Memory networks. We adopt the problem setting presented in (Thickstun, 2021). In the transformer model, an initial embedding layer maps input tokens to $d-$dimensional vector representations. These embeddings then pass through a series of layers, each performing multi-head attention, normalization and feed-forward operations. The input to the $l-$th layer of the transformer is a matrix $\boldsymbol{X}^{(l)} \in \mathbb{R}^{N \times d}$, where $N$ is the number of queries, represented in a $d-$dimensional feature space. A vanilla transformer layer with $H$ attention heads is then defined as follows:

$$\boldsymbol{X}^{(l+1)} = \texttt{LayerNorm}\left(\boldsymbol{Y}^{(l)} + \texttt{MLP}\left(\boldsymbol{Y}^{(l)}\right)\right) \tag{1}$$

$$\boldsymbol{Y}^{(l)} = \texttt{LayerNorm}\left(\boldsymbol{X}^{(l)} + \texttt{Attn}\left(\boldsymbol{X}^{(l)}\right)\right) \tag{2}$$

$$\texttt{Attn}\left(\boldsymbol{X}^{(l)}\right) = \boldsymbol{X}^{(l)} + \sum_{h=1}^{H} \texttt{softmax}\left(\frac{\boldsymbol{X}^{(l)}\boldsymbol{Q}_h^{(l)}\boldsymbol{K}_h^{(l)^T}\boldsymbol{X}^{(l)^T}}{\sqrt{d}}\right)\boldsymbol{X}^{(l)}\boldsymbol{V}_h^{(l)}\boldsymbol{P}_h^{(l)^T} \tag{3}$$

$$\texttt{MLP}\left(\boldsymbol{X}^{(l)}\right) = \texttt{ReLU}\left(\boldsymbol{X}^{(l)}\boldsymbol{G}_1^T + \mathbf{1}_N\boldsymbol{b}_1^T\right)\boldsymbol{G}_2^T + \mathbf{1}_N\boldsymbol{b}_2^T, \tag{4}$$

where $\boldsymbol{K}_h^{(l)}, \boldsymbol{Q}_h^{(l)}, \boldsymbol{V}_h^{(l)}, \boldsymbol{P}_h^{(l)} \in \mathbb{R}^{d \times d_H}$ are the key, query, value and projection matrices respectively, for head $h$ and layer $l$.

## 2.2 LOW RANK (MATRIX) ADAPTATION

LoRA modifies the pre-trained weights by adding a trainable update. Explicitly, at each layer and head $h$:

$$\boldsymbol{K}_h = \boldsymbol{K}_h^0 + d\boldsymbol{K}_h, \quad \boldsymbol{Q}_h = \boldsymbol{Q}_h^0 + d\boldsymbol{Q}_h, \quad \boldsymbol{V}_h = \boldsymbol{V}_h^0 + d\boldsymbol{V}_h, \quad \boldsymbol{P}_h = \boldsymbol{P}_h^0 + d\boldsymbol{P}_h, \qquad (5)$$

where $\boldsymbol{K}^0, \boldsymbol{Q}^0, \boldsymbol{V}^0, \boldsymbol{P}^0$ denote the pre-trained weights and $d\boldsymbol{K}, d\boldsymbol{Q}, d\boldsymbol{V}, d\boldsymbol{P}$ the trainable adapters.

While each attention head's MLP contains two trainable matrices, $\boldsymbol{G}_1$ and $\boldsymbol{G}_2$, our focus is on fine-tuning the attention matrices. This has been demonstrated to be effective for LLM adaptation (Hu et al., 2021; Kopiczko et al., 2023). Nevertheless, these methods can be easily extended to other parameters, including the MLP weights.

Let $\boldsymbol{W}_h \in \{\boldsymbol{Q}_h, \boldsymbol{K}_h, \boldsymbol{V}_h, \boldsymbol{P}_h\}$ for $h = 1, \ldots, H$ denote the query, key, value and projection matrices, respectively, for each attention head. After concatenating updates across all attention heads, we get:

$$d\tilde{\boldsymbol{W}} = (d\boldsymbol{W}_1, \ldots, d\boldsymbol{W}_H) \in \mathbb{R}^{d \times d}.$$

Hu et al. (2021) proposed to parametrize the updates using rank-$r$ matrices, which can be expressed as

$$d\tilde{\boldsymbol{W}} = \frac{\alpha}{r} \boldsymbol{A} \boldsymbol{B}^T, \quad \boldsymbol{A}, \boldsymbol{B} \in \mathbb{R}^{d \times r}, \qquad (6)$$

where $\alpha$ is a constant and $r$ denotes the rank of the update. The scaling factor simply aims to reduce the efforts of re-tuning the learning rate when training adapters of varying rank. It has been shown that while this scaling heuristic works well for smaller ranks, it can be sub-optimal for larger ranks (Kalajdzievski, 2023). Hayou et al. (2024) have also shown that setting the learning rate for the $\boldsymbol{A}$ and $\boldsymbol{B}$ matrices appropriately can further improve convergence and performance.

Although LoRA is an efficient fine-tuning technique, the number of parameters required for each layer is at least $8 \cdot d \cdot r$. Thus, the total number of trainable parameters is:

$$\#\text{parameters (LoRA)} = 8 \cdot d \cdot L \cdot r, \qquad (7)$$

where $L$ is the total number of layers. Even with $r = 1$, this results in $8 \cdot d \cdot L$ parameters. In practice, for LLMs with high dimensionality ($d$) and many layers ($L$), this lower bound can still lead to a significant number of trainable parameters.

LoRA has also been combined with model weight quantization (Dettmers et al., 2024), further decreasing resource requirements. Unlike adapter-based PEFT methods (Houlsby et al., 2019; Pfeiffer et al., 2020; Rücklé et al., 2020; He et al., 2021), LoRA does not introduce additional inference time overhead during deployment, as the trainable matrices can be integrated with the fixed weights.

Building upon this foundation, AdaLoRA (Zhang et al., 2023b) expands the LoRA technique by introducing dynamic rank adjustment for low-rank matrices during fine-tuning. The fundamental concept involves optimally allocating the parameter resources by selectively pruning less crucial components of the matrices based on an importance metric. LoRA-FA (Zhang et al., 2023a) reduces the number of trainable parameters by freezing the $\boldsymbol{A}$ matrix to its random initialisation, while achieving similar performance to LoRA.

## 2.3 TENSOR ALGEBRA

In the following sections we introduce our proposed LoRTA framework, which is a tensor adaptation model for PEFT. To facilitate the upcoming analysis, we briefly present some tensor algebra preliminaries and refer the reader to Appendix A and Sidiropoulos et al. (2017); Kolda & Bader (2009) for further details.

A $N$-order tensor $\mathcal{X} \in \mathbb{R}^{I_1 \times I_2 \times \cdots \times I_N}$ is an $N$-way array indexed by $i_1, i_2, \ldots, i_N$ with elements $\mathcal{X}(i_1, i_2, \ldots, i_N)$. It consists of $N$ types of modes: $\mathcal{X}(:, i_2, \ldots, i_N)$, $\mathcal{X}(i_1, :, \ldots, i_N), \ldots, \mathcal{X}(i_1, i_2, \ldots, :)$. Any tensor can be decomposed as a sum of $N$-way outer products as:

$$\mathcal{X} = \sum_{r=1}^{R} \boldsymbol{A}_1[:, r] \circ \boldsymbol{A}_2[:, r] \circ \cdots \circ \boldsymbol{A}_N[:, r], \qquad (8)$$

where $\boldsymbol{A_n} = [\boldsymbol{a}_n^1, \boldsymbol{a}_n^2, \ldots, \boldsymbol{a}_n^R] \in \mathbb{R}^{I_n \times R}$, $n = 1, \ldots, N$ are called the low rank factors of the tensor. The above expression represents the *canonical polyadic decomposition* (CPD) or *parallel factor analysis* (PARAFAC) (Harshman & Lundy, 1994) of a tensor. A tensor can be fully characterized by its latent factors, so we can represent a tensor by its CPD model as:

$$\mathcal{X} = [\![\boldsymbol{A}_1, \boldsymbol{A}_2, \ldots, \boldsymbol{A}_N]\!]. \tag{9}$$

Unlike other tensor models, such as Tucker and Block Term Decomposition (BTD), the CPD model is unique under certain conditions. As a result, the CPD model is often preferred when the goal is to minimize the number of parameters.

A tensor can also be represented as a set of matrices, by fixing all the modes but two as:

$$\mathcal{X}[:,:,i_3, \ldots, i_N] = \boldsymbol{A}_1 \left( \mathrm{Diag}\left(\boldsymbol{A}_3\left(i_3,:\right)\right) \odot \cdots \odot \mathrm{Diag}\left(\boldsymbol{A}_N\left(i_N,:\right)\right) \right) \boldsymbol{A}_2^T, \tag{10}$$

where $\mathrm{Diag}\left(\boldsymbol{A}_n\left(i_n,:\right)\right)$ is the diagonal matrix with diagonal equal to $\boldsymbol{A}_N\left(i_n,:\right)$.

## 3 LOW RANK TENSOR ADAPTATION

### 3.1 PARAMETER SHARING ACROSS LAYERS

To further increase the compression ratio in PEFT models, recent works (Kopiczko et al., 2023; Koohpayegani et al., 2024) suggest sharing parameters across layers that operate as predefined projection matrices. As we see next, this leads to tensor factorization models with fixed parameters.

**Vector-based Random Matrix Adaptation (VeRA)** Kopiczko et al. (2023) have proposed to parameterize updates using two learnable vectors at each layer and fixed random matrices shared across all layers. The update at each layer can be expressed as

$$d\tilde{\boldsymbol{W}} = \boldsymbol{A}\mathrm{Diag}\left(\boldsymbol{C}_D[l,:]\right)\boldsymbol{B}^T\mathrm{Diag}\left(\boldsymbol{C}_B[l,:]\right), \tag{11}$$

where $\boldsymbol{A}, \boldsymbol{B} \in \mathbb{R}^{d \times r}$ are the random projections, and $\boldsymbol{C}_D \in \mathbb{R}^{L \times r}$, $\boldsymbol{C}_B \in \mathbb{R}^{L \times d}$ are matrices that collect trainable vectors across layers. The model in 11 is a coupled matrix factorization model and is similar to a tensor model. In particular, if we remove the $\boldsymbol{C}_B$ term VeRA can be interpreted as a low-rank tensor CPD parameterization with fixed random factors. That is, the weight update $\tilde{\boldsymbol{W}}$ is a rank-$r$ third order tensor $\mathcal{T} \in \mathbb{R}^{d \times d \times L}$. Note that, omitting the $C_B$ term has been shown to lead to a small performance degradation unlike omitting $C_D$ (Kopiczko et al., 2023).

**Random Matrix basis Adaptation (NOLA)** In a similar manner, Koohpayegani et al. (2024) have proposed to parameterize the weight update by expressing the matrices $\boldsymbol{A}$ and $\boldsymbol{B}$ as linear combinations of fixed random basis matrices, that are shared across all layers. The weight update $d\boldsymbol{W}$ for layer $l$ is then given by:

$$d\tilde{\boldsymbol{W}}_l = \sum_{i=1}^{k}\sum_{j=1}^{k}\alpha_{(i,l)}\beta_{(j,l)}\boldsymbol{A}_i\boldsymbol{B}_j^T, \tag{12}$$

where $\boldsymbol{A}_i, \boldsymbol{B}_j \in \mathbb{R}^{d \times r}$ are fixed random matrices, shared across all layers, and $\boldsymbol{\alpha}_l = \left\{\alpha_{(i,l)}\right\}_{i=1}^{K}$ and $\boldsymbol{\beta}_l = \left\{\beta_{(i,l)}\right\}_{i=1}^{K}$ are the learned coefficients for each layer. If we stack the random matrices $\boldsymbol{A}_i, \boldsymbol{B}_j \in \mathbb{R}^{d \times r}$ into tensors $\mathcal{A}$, $\mathcal{B}$ such that: $\mathcal{A}[:,:,i] = \boldsymbol{A}_i$ and $\mathcal{B}[:,:,j] = \boldsymbol{B}_j$, then 12 can be cast as:

$$d\tilde{\boldsymbol{W}}_l = \sum_{i=1}^{k}\sum_{j=1}^{k}\alpha_{(i,l)}\beta_{(j,l)}\sum_{m=1}^{r}\mathcal{A}[:,m,i]\mathcal{B}[:,m,j]^T = \sum_{m=1}^{r}\mathcal{A}[:,m,:]\left(\boldsymbol{\alpha}_l\boldsymbol{\beta}_l^T\right)\mathcal{B}[:,m,:]^T, \tag{13}$$

and $d\tilde{\boldsymbol{W}}_l$ admits the following factorization. $d\tilde{\boldsymbol{W}}_l = \sum_{m=1}^{r}\boldsymbol{P}_A^{(m)}\left(\boldsymbol{\alpha}_l\boldsymbol{\beta}_l^T\right)\boldsymbol{P}_B^{(m)T}$, where $\boldsymbol{P}_A^{(m)} = \mathcal{A}[:,m,:]$, and $\boldsymbol{P}_B^{(m)} = \mathcal{B}[:,m,:]$ are also random projection matrices with different dimensions compared to $\boldsymbol{A}_i$, $\boldsymbol{B}_j$. As a result, NOLA can be viewed as the following tensor factorization model:

$$d\tilde{\mathcal{W}} = \sum_{m=1}^{r}\left[\!\left[\boldsymbol{P}_A^{(m)}\tilde{\boldsymbol{A}}, \boldsymbol{P}_B^{(m)}\tilde{\boldsymbol{B}}, \boldsymbol{I}\right]\!\right], \tilde{\boldsymbol{A}}[:,l] = \boldsymbol{\alpha}_l, \tilde{\boldsymbol{B}}[:,l] = \boldsymbol{\beta}_l. \tag{14}$$

The expression in 14 is a a summation of CPD models, also known as Block Term Decomposition, which is an expressive tensor model, but can lack parsimony (Kolda & Bader, 2009).

### 3.2 LoRTA: A more efficient tensor model

In the previous section, we explored PEFT models that share parameters across layers, highlighting their correspondence to tensor factorization models. Namely, VeRA and NOLA utilize fixed projection matrices shared across layers. However, this strategy can result in models that are larger than necessary relative to their degrees of freedom due to the inclusion of these random matrices. Although these matrices can be generated on the fly by solely storing the pseudo-random number generator seed, this still incurs additional resource demands during training, and increases loading time for inference.

To address this issue, we propose modeling the trainable adapters using a low-rank CPD structure. This choice is motivated by the favorable properties of CPD: it is universal, capable of exactly factorizing any tensor, yet remains concise and parsimonious, typically requiring only a small number of parameters to achieve low approximation error (Sidiropoulos et al., 2017). This contrasts with tensor adapters used in vision (Jie & Deng, 2023) and recently in LLM finetuning (Bershatsky et al., 2024), which rely on Tucker and Tensor-Train models. In fact, for small ranks, CPD is equivalent to Tucker when the core tensor in Tucker is the identity tensor. However Tucker is always parametrized with a dense tensor and therefore requires a larger number of parameters for the same rank.

LoRTA represents all weight updates as a 5th-order tensor $d\tilde{\mathcal{W}} \in \mathbb{R}^{d \times \frac{d}{H} \times H \times L \times M}$. By integrating updates of layers, heads and the $\boldsymbol{Q}$, $\boldsymbol{K}$, $\boldsymbol{V}$, $\boldsymbol{P}$ matrices into a unified low-rank CPD tensor model, LoRTA exploits redundancy across different modes of the tensor. This approach can thus not only improve parameter efficiency but also facilitate learning by exploiting the shared information among various components of the model. This contrasts with existing PEFT approaches, which tensorize each weight update independently (Yang et al., 2024) or only share parameters across layers (Jie & Deng, 2023; Bershatsky et al., 2024). In order to illustrate how additional tensor modes can result in parameter efficiency gains, Figure 1 compares – for a single weight update – LoRA with a rank one tensor model that adds attention heads as a mode.

By utilizing a low-rank CPD model, we can express this tensor as:

$$d\tilde{\mathcal{W}} = [\![\boldsymbol{A}, \boldsymbol{B}, \boldsymbol{C}_H, \boldsymbol{C}_L, \boldsymbol{C}_M]\!],$$

where $\boldsymbol{A} \in \mathbb{R}^{d \times r}$ and $\boldsymbol{B} \in \mathbb{R}^{\frac{d}{H} \times r}$ are factor matrices for the input and output dimensions, respectively, and $\boldsymbol{C}_H \in \mathbb{R}^{H \times r}$, $\boldsymbol{C}_L \in \mathbb{R}^{L \times r}$, $\boldsymbol{C}_M \in \mathbb{R}^{4 \times r}$ are factor matrices for the attention heads, layers, and the four matrices $Q$, $K$, $V$, $P$. Each weight matrix update can then be retrieved as:

$$d\tilde{\mathcal{W}}[:, :, k, l, i] = \boldsymbol{A} \left( \text{Diag}\left(\boldsymbol{C}_H[k, :]\right) \text{Diag}\left(\boldsymbol{C}_L[l, :]\right) \text{Diag}\left(\boldsymbol{C}_M[i, :]\right) \right) \boldsymbol{B}^\top,$$

where $k$ indexes the attention heads, $l$ indexes the layers, and $i$ indexes the matrices $\boldsymbol{Q}$, $\boldsymbol{K}$, $\boldsymbol{V}$, $\boldsymbol{P}$. Note that, unlike previous implicit tensor models such as NOLA and VeRA, which rely on fixed random projections or parameters and learn only scaling coefficients, our proposed model is *fully trainable*. All factor matrices ($\boldsymbol{A}$, $\boldsymbol{B}$, $\boldsymbol{C}_H$, $\boldsymbol{C}_L$, $\boldsymbol{C}_M$) are learned during training, providing greater expressiveness and forgoing the dependency on pre-defined random bases or projections.

Table 1 shows how the CP low rank tensor parameterization leads to better scaling in the number of parameters with respect to the tensor rank $r$. Moreover, our higher-order weight update tensorization improves scaling in terms of transformer architecture hyperparameters, namely the embedding dimension $d$, number of attention heads $H$, and number of fine-tuned attention matrices $M$.

### 3.3 Other Low Rank Tensor models in PEFT

As mentioned in the previous section, existing PEFT tensor-based models differ from our method both in their parameter-sharing schemes, which result from different weight update tensorization approaches, as well as in the low-rank tensor models they employ. Below, we provide a concise overview of these approaches which intends to highlight the differences with LoRTA; further details are available in Appendix X and the provided references.

**FaCT & LoTR** In the context of vision transformers, Jie & Deng (2023) have proposed to represent updates across all layers as a third order tensor $d\tilde{\mathcal{W}} \in \mathbb{R}^{L \times d \times d}$. They propose two parameterizations of $d\tilde{\mathcal{W}}$, namely, a Tensor Train and Tucker3 low rank tensor models. Recently, Bershatsky et al.

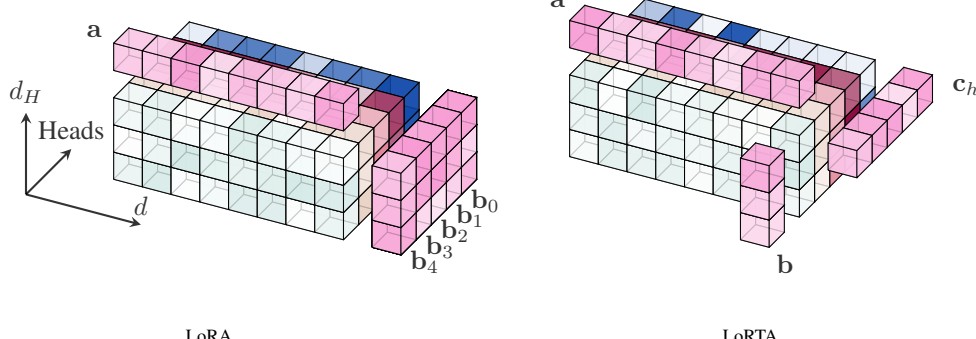

Figure 1. Illustration of a rank 1 adapter for a single weight matrix with multiple heads. (Left) The LoRA update for head $h$ is computed as $d\boldsymbol{W}_h = \boldsymbol{b}_h \circ \boldsymbol{a}$. (Right) The update using a third order low rank tensor model is computed as $dW_h = \boldsymbol{b} \circ \boldsymbol{c}[h] \circ \boldsymbol{a}$. Both models have the same tensor rank, but the latter has less parameters.

(2024) have proposed to apply the same tensorization across layers to fine-tune LLMs, but using a low rank Tucker2 tensor model to parameterize updates.

**LoreTTA** Yang et al. (2024) propose two methods that employ low rank tensor models. However, these models do not share parameters across layers, they reparameterize low rank matrix adapters using low rank tensor models. In LoreTTA-rep a low rank matrix model is first applied to each weight update in the same manner as described for LoRA in Equation (6). Then each of the $ML$ resulting LoRA factors $\boldsymbol{A}, \boldsymbol{B} \in \mathbb{R}^{d \times r}$ are expressed as a n-th order tensor with arbitrary dimensions, i.e. $\mathcal{A}, \mathcal{B} \in \mathbb{R}^{k_1 \times \ldots \times k_N}$. Finally, each of these tensors is parametrized Tensor Train model, explicitly, $\mathcal{A} = \prod_{i=1} \boldsymbol{G}_i$ where $\boldsymbol{G}_i \in \mathbb{R}^{r \times k_i \times r}$. We highlight that the added dimensions $k_i$ are hyperparameters that must satisfy $\prod_i k_i = dr$ and $k_i \geq r$ for all $i$; otherwise, it would induce a new tensor rank deficiency. Moreover, choosing appropriate values of $k_i$ might be challenging and necesitate further hyperparameter tuning. Yang et al. (2024) also proposed LoReTTA-adp, applying a tucker parameterization to an adapter method, which unlike our method and the rest of the aforementioned methods adds new parameters to the model and thus can not be merged into the original weights, thereby incurring additional inference costs.

## 4 EXPERIMENTS

### 4.1 NATURAL LANGUAGE UNDERSTANDING

We evaluate our approach by fine-tuning RoBERTa models on the General Language Understanding Evaluation (GLUE) Wang et al. (2018) benchmark. We conduct experiments across three distinct settings previously reported in the literature by Bershatsky et al. (2024), Yang et al. (2024) and Kopiczko et al. (2023). These settings differ in hyperparameters, including the number of training

| Method | Update Tensor shape | Tensor Model | Parameters | r=4 | r=64 |
|---|---|---|---|---|---|
| LoRA | $ML \times d \times d$ | Matrix-Batch | $2MLdr$ | 2.1M | 33M |
| LoReTTA | $ML \times k_1 \times \ldots \times k_6$ | Custom | $MLr^2 \sum_i k_i$ | 92k | 50M |
| LoTR | $ML \times d \times d$ | Tucker2 | $MLr^2 + 2dr$ | 33k | 786k |
| FacT-TT | $ML \times d \times d$ | Tensor-Train | $MLr^2 + 2dr$ | 33k | 786k |
| FacT-TK | $ML \times d \times d$ | Tucker3 | $(2d + ML)r + r^3$ | 33k | 790k |
| Ours | $M \times L \times d \times d/h \times h$ | CP | $(d + d/h + h + L + M)r$ | 17k | 274k |

Table 1: Number of parameters of different Tensor based PEFT methods as a function of the number of finetuned attention/projection matrices $M$, the number of layers, $L$, the embedding dimension $d$, the number of heads $h$ and the tensor rank of the update, $r$. For LoreTTA, $k_i$ are hyperparameters that must satisfy $\prod_i k_i = dr$ and $k_i \geq r$ for all $i$. We also include the number of parameters for the Llama2-7b architecture when finetuning only M=2 attention matrices (e.g. Q and V) for different ranks. For LoReTTa we use $k_1 = \ldots = k_6 = 5$ for $r = 4$ and $k_1 = k_2 = k_3 = 64$ for $r = 64$.

epochs, different learning rates for the classification head and encoder, the learning rate decay strategy (linear vs fixed), the use of different scaling parameters $\alpha$, and the grid search ranges. Because best results on the validation set are reported, performance for the same baseline method can vary considerably across settings (see, for example, LoRA performance reported by Hu et al. (2021), Yang et al. (2024) and Bershatsky et al. (2024)). Therefore, we provide an evaluation of our method in a variety of experimental conditions, while also maintaining the original configurations in which state-of-the-art methods were originally reported. Detailed settings can be found in Table **??** in Appendix E.1.

We also finetuned Llama2 models (Touvron et al., 2023) on question-answering (QA) tasks SQuAD (Rajpurkar et al., 2016), DROP (Dua et al., 2019), COPA (Roemmele et al., 2011), and ReCoRD (Zhang et al., 2018), following the experimental setting outlined by Yang et al. (2024) For these tasks, we used a randomly selected subset of 1,000 samples to simulate a low-data regime and increase the task difficulty. All classification tasks are tackled as language modeling tasks following the prompt-based fine-tuning approach described by Malladi et al. (2023).

**Baselines** We benchmark our method against the following methods:

- **Full finetuning**: all parameters are trained.
- **IA3** (Liu et al., 2022): rescales activations with learned vectors
- **Prefix** (Li & Liang, 2021b): prepends learnable continuous vectors (prefixes) to the input embeddings.
- **LoRA** (Hu et al., 2021), **LoRA-FA** (Zhang et al., 2023a) and **VeRA** (Kopiczko et al., 2023), **LoTR** (Bershatsky et al., 2024), **LoReTTA** (Yang et al., 2024): As previously described.
- We omit **Adapter$^{H}$** (Houlsby et al., 2019), **Adapter$^{P}$** (Pfeiffer et al., 2020), **Bitfit** (Zaken et al., 2021), **AdapterDrop** (Rücklé et al., 2020), and other methods that are customarily reported but have been outperformed by more recent methods in these settings.

The results in Table 2 show that LoRTA can achieve comparable or slightly superior performance with less trainable parameters compared to state of the art tensor based PEFT methods LoreTTA (Yang et al., 2024) and LoTR (Bershatsky et al., 2024) when finetuning RoBERTA base on GLUE tasks. Similarly, for RoBERTa large LoRTA can also achieve a 6x reduction in the number of trainable parameters with only small drop in average performance (2%) when compared to Kopiczko et al. (2023). In this settings we did not tune the hyperparameters for our method as extensively as baselines, and thus this gap could be further reduced.

In Llama QA experiments, shown in Table 3, full fine-tuning (FT) achieves the highest average score (77.3) with 7 billion trainable parameters, but among the PEFT methods LoRTA (r=8) achieves the highest average score (76.7) with just 0.03 million parameters, representing a 17x reduction in parameter count with respect to the most efficient method.

### 4.2 INSTRUCTION TUNING

We fine-tune the 7 billion parameter Llama2 (Touvron et al., 2023) models on the cleaned Alpaca instruction tuning dataset (Taori et al., 2023). We train for one epoch, preceded by a warm-up learning rate sweep as in the standard setting. Other hyperparameters are detailed in Appendix E.2.

As shown in Figure 2, LoRTA effectively reduces the number of parameters to a fraction of those required by the lowest rank in LoRA, with only a small performance cost. The loss decreases monotonically with the number of parameters used, both in training and testing, and LoRTA even demonstrates superior performance with fewer parameters for ranks 96 and 192. To further evaluate the fine-tuned models, we use MT-Bench (Zheng et al., 2023), an LLM-as-a-judge benchmark. MT-Bench assesses multi-turn conversational and instruction-following abilities on 80 open-ended questions, covering diverse capabilities such as roleplaying, reasoning, coding and information retrieval. GPT-4 is used to score the outputs of the model on a scale of one to ten.

As shown in Figure 3, LoRTA can almost match average performance despite using just 1/5th of the parameters (r=48). Unlike the loss observed in the Alpaca dataset, performance does not increase monotonically, potentially due to overfitting. Moreover, performance varies across tasks. For example, most LoRTA models surpass LoRA in reasoning but fall short in writing.

| | Method | # Trainable Parameters | SST-2 | MRPC | CoLA | QNLI | RTE | STS-B | Avg. |
|---|---|---|---|---|---|---|---|---|---|
| LoReTTA | LoRA (r=8) | 630k | 94.01 | 91.48 | 62.08 | 92.39 | 74.51 | 84.69 | 83.19 |
| | LoReTTA rep | 70k | 94.28 | 90.63 | 61.72 | 92.40 | 74.42 | 89.24 | 83.78 |
| | LoRTA (r=20) | 48k | 94.27 | 92.04 | 63.35 | 91.48 | 75.09 | 89.82 | **84.34** |
| | LoRTA (r=12) | 29k | 93.81 | 91.13 | 61.40 | 92.04 | 74.73 | 89.64 | 83.79 |
| LoTR | LoRA (r=8) | 300k | 94.2 | 88.0 | 61.1 | 91.3 | 73.0 | 90.7 | 83.05 |
| | LoTR | 74k | 93.0 | 85.9 | 60.5 | 90.0 | 66.0 | 91.9 | 81.22 |
| | LoRTA (r=16) | 15k | 94.73 | 90.44 | 64.32 | 92.37 | 76.9 | 90.25 | **84.84** |
| | LoRTA (r=4) | **3.4k** | 94.61 | 89.21 | 60.55 | 90.61 | 76.9 | 89.97 | 83.6 |
| VeRA | LoRA | 800k | 96.2 | 90.2 | 68.2 | **94.8** | 85.2 | **92.3** | **87.8** |
| | LoRA-FA | 3.7M | 96.0 | 90.0 | 68.0 | 94.4 | 86.1 | 92.0 | 87.7 |
| | VeRA | 61k | 96.1 | **90.9** | 68.0 | 94.4 | **85.9** | 91.7 | **87.8** |
| | LoRTA (r=8) | **9k** | 96.3 | 89.5 | 65.1 | 94.3 | 85.6 | 91.1 | 85.7 |

Table 2: Performance of RoBERTa Base and Large models on GLUE tasks under three different experimental settings reported by LoReTTA (Yang et al., 2024), LoTR (Bershatsky et al., 2024), and VeRA (Kopiczko et al., 2023). In LoReTTA, LoRTA is applied to the encoder and LoRA to the classifier with the same rank, while for LoTR and VeRA, LoRTA is applied only to the encoder. Trainable parameters include the classifier for LoReTTA but exclude it for LoTR and VeRA, where it is fully trained. VeRA results use RoBERTa Large, whereas LoTR and LoReTTA use RoBERTa Base

| Method | # Trainable Parameters | COPA | ReCoRD | SQuAD | DROP | Avg. |
|---|---|---|---|---|---|---|
| FT | 7B | 86 | 81.1 | 90.71 | 51.38 | 77.3 |
| LoRA (r=8) | 4.19M | 81 | 79.4 | 90.56 | 45.96 | 74.2 |
| Prefix | 1.31M | 83 | 81.0 | 90.56 | 45.95 | 75.1 |
| IA3 | 0.60M | 80 | 81.5 | 89.41 | 39.37 | 72.6 |
| LoRETTA rep | 0.51M | 86 | 80.3 | 88.47 | 42.71 | 74.4 |
| LoRTA (r=4) | 0.02M | 87 | 81.1 | 87.4 | 44.04 | 74.9 |
| LoRTA (r=8) | 0.03M | 87 | 81.6 | 88.5 | 49.7 | **76.7** |

Table 3: LLama2-7B performance on SuperGLUE and question-answering tasks (SQuAD, DROP). We follow the experimental setup used by Yang et al. (2024).

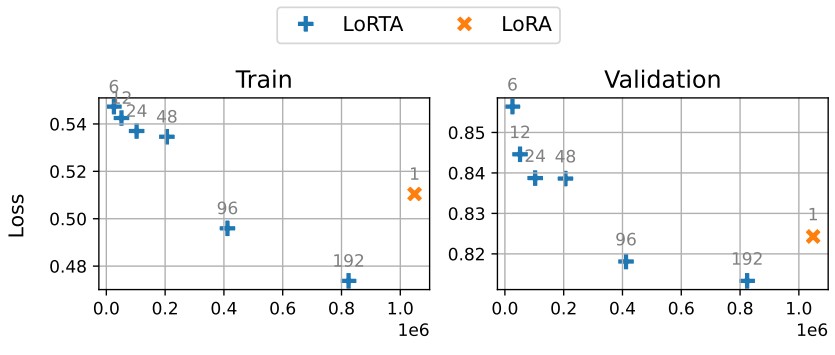

Figure 2. Mean cross-entropy loss on training and testing data for Llama2-7b on the Alpaca dataset vs number of trainable parameters for different adapter ranks. Lower is better. Numbers on top of markers denote the adapter rank.

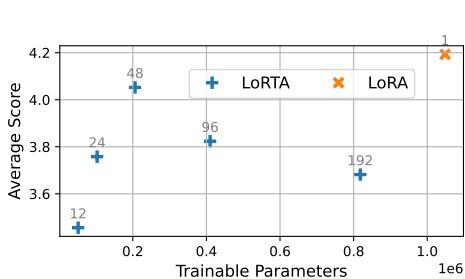 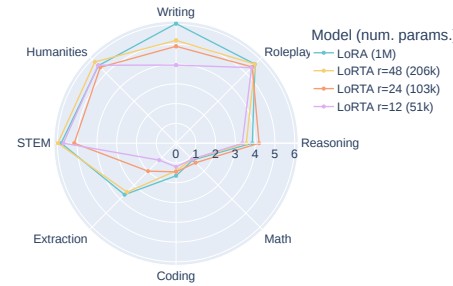

Figure 3. Performance on MT-Bench (Zheng et al., 2023) for Llama2-7b (Touvron et al., 2023) models fine-tuned with LoRA and LoRTA. Higher is better. **Left**: Average score across all questions vs number of trainable parameters. Numbers on top of markers denote the adapter rank. **Right**: Average score per task.

### 4.3 PREFERENCE OPTIMIZATION

While various techniques to align LLMs with human preferences on specific tasks exist (see, for example, Kaufmann et al. (2023) and references therein), we utilize Direct Preference Optimization (DPO) (Rafailov et al., 2024) due to its simplicity and effectiveness.

We fine-tune the 7 billion parameter Llama 2 model (Touvron et al., 2023) on the cleaned version of the Intel Orca dpo pairs dataset[1]. This synthetic preference dataset comprises 6k prompts across various domains and tasks, along with the corresponding outputs from ChatGPT and Llama2-13B. In this version of the dataset, ChatGPT is used to score outputs and the preferred choices are designated based on these scores. Because preference datasets are often small, a KL regularization that penalizes deviations from the pre-trained model's outputs is used to mitigate overfitting. In our experiments, the regularization coefficient $\beta$ was set to $0.1$. We use Huggingface Transformer Reinforcement Learning (trl) library[2]. For a complete description of hyperparameters see Appendix E.3.

As shown in Figure 4 LoRTA exhibited non-monotonic performance across ranks. This suggests that further hyperparameter tuning may be necessary to stabilize its performance. Although we did not tune hyperparameters, most ranks still outperformed LoRA with significantly fewer parameters. We further evaluated the fine-tuned models on the LLM-as-a-judge MT-benchmark. In this setting,

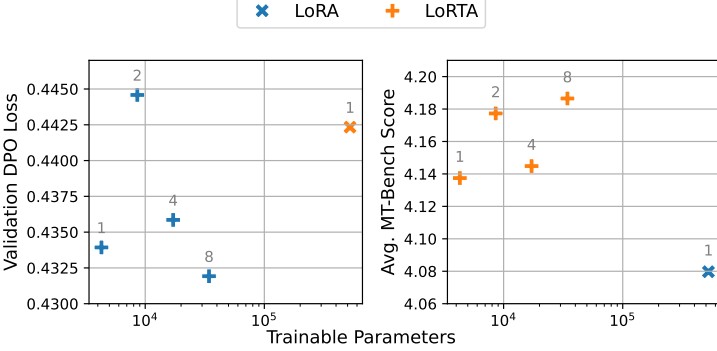

Figure 4. (Left) Mean DPO loss on held-out data from the orca dpo pairs dataset vs number of trainable parameters, lower is better. (Right) MT-Bench average scores Scores vs number of trainable parameters, higher is better.

LoRTA consistently outperformed LoRA across all ranks, including at rank 2 where it had shown higher DPO loss on the preference dataset. This improvement suggests enhanced out-of-distribution generalization capabilities for LoRTA adapters since MT-bench differs from the training dataset.

---

[1]https://huggingface.co/datasets/argilla/distilabel-intel-orca-dpo-pairs
[2]https://github.com/huggingface/trl

### 4.4 PROTEIN FOLDING

Protein folding, the process by which a protein's amino acid sequence determines its three-dimensional structure, is a fundamental problem in molecular biology. Accurate prediction of protein structures from their sequences has significant implications for understanding protein function and designing new proteins for therapeutic purposes. ESMFold (Lin et al., 2023) is a frontier model for this task trained in two stages. First, ESM-2, a BERT-based (Devlin et al., 2019) protein language model, is trained with the masked-language-modeling objective on amino acid sequences. This unsupervised pretraining allows the model to capture complex patterns and relationships within protein sequences. Remarkably, valuable structural information emerges in the model's features during this process (Rao et al., 2020). In the second stage, ESM-2 is frozen, and a model head predicting three-dimensional protein structures is trained on top of language model features.

We re-train ESMFold in the second stage – fine-tuning ESM-2 parameters (we use ESM-2 35M instead of the ESM-2 3B model used in Lin et al. (2023) due to compute constraints) with LoRA and LoRTA instead of freezing them. We evaluate performance with the Local Distance Difference Test for C$\alpha$ atoms (LDDT-C$\alpha$) (Mariani et al., 2013) – that measures accuracy of predicted protein structures by comparing the distance between alpha carbons in predicted and true structures. LDDT-C$\alpha$ ranges from 0 (poor accuracy) to 1 (perfect match). See Appendix E.4 for experiment details.

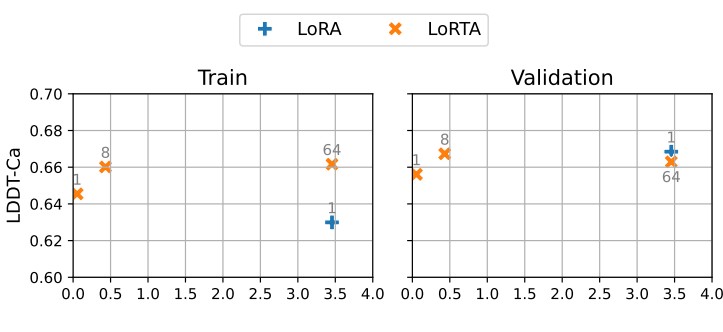

Figure 5. Mean LDDT-C$\alpha$ on train and held-out test sets. Higher is better. LoRTA rank 1 is competitive with LoRA rank 1 on the test set despite having 64x fewer parameters. Numbers on top of markers denote the adapter rank.

As shown in Figure 5, all tested LoRTA ranks outperform rank 1 LoRA on the training set; on the validation set, all tested LoRTA ranks are competitive with rank 1 LoRA. Notably, rank 1 LoRTA is competitive with rank 1 LoRA despite having an order of magnitude fewer parameters.

## 5 CONCLUSION

We have introduced LoRTA, a novel approach that employs a low-rank tensor model for LLM updates. By extending low-rank adaptation to higher-order tensors, LoRTA overcomes the inherent lower bounds on the number of trainable parameters while offering finer-grained control over adapter size. Our experiments across various benchmarks demonstrate that LoRTA achieves comparable and sometimes superior performance than baselines at a reduced parameter count.

Furthermore, we have shown that previous works have implicitly utilized low-rank tensor models with random factors. Nothing precludes our higher-order tensor model from using randomized factors for increased efficiency—a potential direction for future work that could further reduce computational overhead. Lastly, developing more efficient implementations of tensor operations that result in greater memory efficiency also remains a relevant future work direction which could make LoRTA even more suitable for resource-constrained environments.

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

## A  TENSOR ALGEBRA

To facilitate our analysis, we briefly present some tensor algebra preliminaries and refer the reader to Sidiropoulos et al. (2017); Kolda & Bader (2009) for further details.

A $N$-order tensor $\mathcal{X} \in \mathbb{R}^{I_1 \times I_2 \times \cdots \times I_N}$ is an $N$-way array indexed by $i_1, i_2, \ldots, i_N$ with elements $\mathcal{X}(i_1, i_2, \ldots, i_N)$. It consists of $N$ types of modes: $\mathcal{X}(:, i_2, \ldots, i_N)$, $\mathcal{X}(i_1, :, \ldots, i_N), \ldots, \mathcal{X}(i_1, i_2, \ldots, :)$.

A rank-one tensor $\mathcal{Z} \in \mathbb{R}^{I_1 \times I_2 \times \cdots \times I_N}$ is the outer product of $N$ vectors defined as:

$$\mathcal{Z} = \boldsymbol{a}_1 \circ \boldsymbol{a}_2 \circ \cdots \circ \boldsymbol{a}_N, \tag{15}$$

where $\boldsymbol{a}_1 \in \mathbb{R}^{I_1}$, $\boldsymbol{a}_2 \in \mathbb{R}^{I_2}, \ldots, \boldsymbol{a}_N \in \mathbb{R}^{I_N}$ and $\circ$ denotes the outer product. The elementwise formula of the above expression is:

$$\mathcal{Z}(i_1, i_2, \ldots, i_N) = \boldsymbol{a}_1(i_1)\boldsymbol{a}_2(i_2) \cdots \boldsymbol{a}_N(i_N), \quad \text{for all} i_1, i_2, \ldots, i_N, \tag{16}$$

Any tensor can be realized as a sum of $N$-way outer products (rank one tensors), i.e.

$$\mathcal{X} = \sum_{r=1}^{R} \boldsymbol{a}_1^f \circ \boldsymbol{a}_2^f \circ \cdots \circ \boldsymbol{a}_N^f. \tag{17}$$

The above expression represents the *canonical polyadic decomposition* (CPD) or *parallel factor analysis* (PARAFAC) (Harshman & Lundy, 1994) of a tensor. The CPD elementwise representation is:

$$\mathcal{X}(i, j, k) = \sum_{r=1}^{R} \boldsymbol{A}_1(i_1, f)\boldsymbol{A}_2(i_2, f) \cdots \boldsymbol{A}_N(i_N, f), \tag{18}$$

where $\boldsymbol{A_n} = [\boldsymbol{a}_n^1, \boldsymbol{a}_n^2, \ldots, \boldsymbol{a}_n^F] \in \mathbb{R}^{I_n \times F}$, $n = 1, \ldots, N$ are called the low rank factors of the tensor. A tensor can be fully characterized by its latent factors, so we can represent a tensor by its CPD model as:

$$\mathcal{X} = [\![\boldsymbol{A}_1, \boldsymbol{A}_2, \ldots, \boldsymbol{A}_N]\!]. \tag{19}$$

A tensor can be also represented as a set of matrices, by fixing all the modes but two as:

$$\mathcal{X}[:, :, i_3, \ldots, i_N] =$$
$$\boldsymbol{A}_1 \left(\text{Diag}\left(\boldsymbol{A}_3\left(i_3, :\right)\right) \odot \cdots \odot \text{Diag}\left(\boldsymbol{A}_N\left(i_N, :\right)\right)\right) \boldsymbol{A}_2^T, \tag{20}$$

where $\text{Diag}\left(\boldsymbol{A}_n\left(i_n, :\right)\right)$ is the diagonal matrix with diagonal equal to $\boldsymbol{A}_N\left(i_n, :\right)$.

## B  ADDITIONAL RELATED WORK

**Model Compression** While these techniques differ from PEFT in that they focus on reducing the requirements of a trained model rather than efficient adaptation, they offer valuable insights for developing more efficient PEFT approaches. Pruning and quantization are key techniques for compressing neural networks, that have also been extensively applied to LLMs. Pruning removes less important weights, with some methods achieving high compression rates, e.g. (Ma et al., 2023). Quantization reduces weight precision, decreasing model size and also allowing more efficient operations (Lin et al., 2024a). Knowledge distillation is an alternative approach that involves transferring knowledge from a large "teacher" model to a smaller "student" model (Gu et al., 2024).

**Low Rank Training.** Exploiting low rank structure to improve efficiency during both training and inference in deep models has long been studied (Sainath et al., 2013), and also combined with sparsity (Sprechmann et al., 2015). Recent advancements include Cuttlefish (Wang et al., 2023) and ELRT (Sui et al., 2024).

**Data efficient fine tuning.** An alternative approach to reducing fine-tuning costs is to reduce the amount of data. In this direction, Few-shot and continual learning approaches have been shown to be effective in LLM fine-tuning tasks (Lin et al., 2024b; Wang et al., 2024).

**Efficient Architectures** Another relevant direction in resource usage is using more efficient model architectures. Mixture of Experts (MoE) technique, implemented in models like Switch Transformers (Fedus et al., 2022) and GLaM (Du et al., 2022), has shown promise in scaling model capacity

while maintaining computational efficiency by activating only relevant sub-models for given inputs. There is also relevant work on non-transformer architectures, such as RWKV (Peng et al., 2023) and Mamba (Gu & Dao, 2023), which combines the strengths of RNNs and Transformers to achieve efficient inference and training.

## C    OTHER TENSOR LOW RANK MODELS IN PEFT

## D    PARAMETER EFFICIENCY COMPARISON AGAINST LORA.

To fairly compare the parameter efficiency of LoRTA with LoRA, we adjust the tensor rank in LoRTA to match the effective total tensor rank in LoRA, which is $r' = r \times 4L$ due to LoRA applying a rank $r$ update to each of the $4L$ matrices individually. For a given tensor rank, LoRTA reduces the number of parameters from scaling $8dLr$ in LoRA to $4L(d(1 + 1/h) + h + L + 4)r$ in LoRTA (usually $d \gg L$ and $d \gg h$), achieving substantial parameter savings without compromising expressive power. For example, this amounts to a $47.6\%$ reduction in a LLaMA2 7B model.

We provide a breakdown of the parameter savings achieved by our proposed method, LoRTA, compared to LoRA, by parameterizing the weight updates using low-rank tensor decompositions at different granularities. The table below summarizes the dimensions of the update tensors, the number of update tensors used, and the corresponding parameter savings when the tensor rank $r$ matches the tensor rank of LoRA rank $r$. The first row corresponds to LoRA.

Table 4: Update Tensor Modes, Parameters, and Savings

| Added Modes | Update Tensor Dimensions | Number of Update Tensors | Parameter Savings |
|---|---|---|---|
| | $d \times d$ | $4L$ | $0$ |
| Heads | $d \times \frac{d}{H} \times H$ | $4L$ | $1 - \frac{d\left(1 + \frac{1}{H}\right) + H}{2dr}$ |
| Heads, QKVP | $d \times \frac{d}{H} \times H \times 4$ | $L$ | $1 - \frac{d\left(1 + \frac{1}{H}\right) + H + 4}{2dr}$ |
| Heads, QKVP, Layers | $d \times \frac{d}{H} \times H \times 4 \times L$ | $1$ | $1 - \frac{d\left(1 + \frac{1}{H}\right) + H + 4 + L}{2dr}$ |

## E    EXPERIMENTAL DETAILS

In this appendix, we provide further details on the experiments presented in the main paper.

### E.1    NLU

In our GLUE experiments we implemented our method using Huggingface's PEFT, VeRA Kopiczko et al. (2023) and LoreTTA Yang et al. (2024) codebases. Hyperparameters for each of the three settings reported are detailed below.

| Hyperparameter | Value |
|---|---|
| $\alpha$ | 16 |
| Learning Rate | [2E-3, 5E-4] |
| Scheduler | Constant |
| Optimizer | AdamW |
| Number of Epochs | 20 |
| Batch Size | [16, 32] |
| Warmup Steps | 500 |

Table 5: Hyperparameter configurations for RoBERTa Base on the GLUE benchmark following the setup reported by Yang et al. (2024), where only the batch size and learning rate are tuned for each task, selecting between two values based on validation performance. All other hyperparameters match those reported by Yang et al. (2024).

| Hyperparameter | Value |
|---|---|
| $\alpha$ | [0.5 1.0 2.0 8.0] |
| Learning Rate | [5e-4, 1e-3, 5e-3, 1e-2] |
| Scheduler | Linear |
| Optimizer | AdamW |
| Number of Epochs | 20 |
| Batch Size | 32 |
| Warmup Ratio | 0.06 |

Table 6: Hyperparameter configurations for RoBERTa Base on the GLUE benchmark following Bershatsky et al. (2024). A grid-search to set the learning rate and scale parameter for each task is conducted across the specified values.

| Hyperparameter | SST-2 | MRPC | CoLA | QNLI | RTE | STS-B |
|---|---|---|---|---|---|---|
| Optimizer | | | AdamW | | | |
| Warmup Ratio | | | 0.06 | | | |
| LR Schedule | | | Linear | | | |
| Epochs | 10 | 40 | 40 | 20 | 40 | 20 |
| Learning Rate (Head) | 6E-3 | 3E-3 | 6E-3 | 2E-4 | 2E-3 | 2E-3 |
| Learning Rate (Encoder) | 1E-2 | 1E-2 | 1E-2 | 1E-2 | 2E-2 | 2E-2 |
| Batch Size | | | 32 | | | |

Table 7: Hyperparameter configurations for RoBERTa large on the GLUE benchmark. All other hyperparameters are taken from Kopiczko et al. (2023).

### E.2 INSTRUCTION TUNING

For instruction tuning experiments we utilized Lightning AI's LitGPT codebase and training recipe. Hyperparameters are detailed below.

| Parameter | Value |
|---|---|
| $\alpha$ | 16 |
| Learning Rate | 0.01 |
| Scheduler | Cosine |
| Optimizer | AdamW |
| Weight Decay | 0.01 |
| Number of Epochs | 1 |
| Steps | 51000 |
| Batch Size | 16 |
| Warmup Steps | 318 |

Table 8: Hyperparameter configurations for LLama2-7B on the Alpaca dataset.

### E.3 DPO

For preference optimization experiments we utilized using Huggingface trl library's dpo implementation and example script. Hyperparameters are detailed below.

### E.4 PROTEIN FOLDING

For protein folding experiments, we utilized OpenFold Ahdritz et al. (2024) training code and datasets. The following modifications were made to the ESMFold model architecture due to limited compute resources: a) utilize 12 Evoformer layers instead of the 48 used in (Lin et al., 2023) b) utilize ESM-2 35M instead of ESM-2 3B c) maintain outer product mean implementation from (Jumper et al., 2021).

Table 9: Hyperparameter configurations for LLama2-7B on intel orca DPO pairs.

| Parameter | Value |
|---|---|
| $\alpha$ | 16 |
| Learning Rate | 0.00005 |
| Scheduler | Cosine |
| Optimizer | AdamW |
| Weight Decay | 0 |
| Number of Epochs | 1 |
| Batch Size | 16 |
| Warmup Steps | 200 |

Optimizer and learning rate scheduler were identical to (Jumper et al., 2021). Models were trained for 850,000 steps with batch size of 32. Validation metrics were computed using the validation set from (Ahdritz et al., 2024).

Preliminary experiments revealed that higher values of $\alpha$ yield better results in this setting. $\alpha$ for LoRA and LoRTA experiments was then selected in multiple stages. Initially, models were trained with $\alpha$ values of $256 \times r$ and $128 \times r$, and the best-performing model was chosen. If both configurations diverged, $\alpha$ was halved, and models were retrained with the next lower pair (e.g., $64 \times r$ and $32 \times r$). This halving process continued until a convergent model was found. See Table 10 for the selected $\alpha$ values across experiments.

Table 10: Selected $\alpha$ and LDDT-CA for protein folding models.

| Model | $\alpha$ | Validation LDDT-C$\alpha$ |
|---|---|---|
| LoRA (r = 1) | 128 | 0.668 |
| LoRTA (r = 64) | 128 | 0.663 |
| LoRTA (r = 8) | 256 | 0.667 |
| LoRTA (r = 1) | 2 | 0.656 |

## F    ADDITIONAL RESULTS

Figure 6 shows that Validation gains were primarily driven by reduced training error, though generalization slightly worsened, particularly at rank 2. On the other hand, as already mentioned, MT-bench performance was comparable o superior for LoRTA across all ranks, as shown in Figure 7.

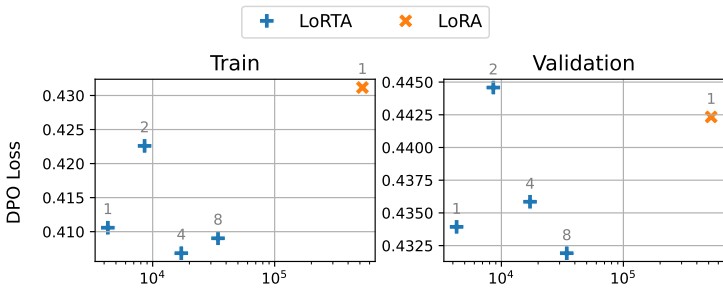

Figure 6. Mean DPO loss on the training (Left) and on held-out data (Right) from the orca dpo pairs dataset vs number of trainable parameters, lower is better.

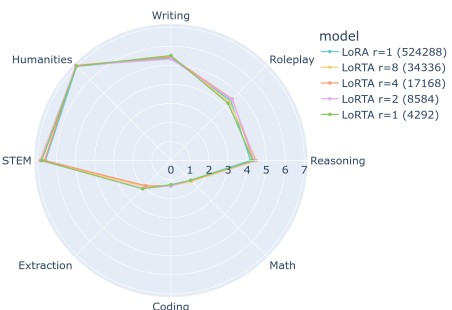

Figure 7. Performance on MT-Bench Zheng et al. (2023) for llama2-7b Touvron et al. (2023) models fine-tuned with LoRA and LoRTA using dpo on intel orca pairs. Average score per task. Higher is better.

## G  PRACTICAL IMPLICATIONS OF ADAPTER SIZE REDUCTION

The reduction in adapter size is primarily motivated by the need to improve task-switching efficiency and minimize storage requirements in scenarios involving a large number—potentially thousands—of adapters. Frequent CPU-GPU transfers for loading adapters in such settings can introduce significant overhead. By further compressing parameters, it becomes feasible for thousands of customized models to coexist with a base LLM in GPU memory, substantially enhancing scalability and performance in multi-task environments.

During training, the reduction in GPU memory usage from shrinking optimizer states is marginal for parameter reductions beyond LoRA. Memory consumption in these cases is dominated by activations and caches stored during forward and backpropagation. Additional memory savings could be achieved by compressing activations or gradients, leveraging the low-rank structure of updates, or dynamically recomputing them. While our model features fewer trainable parameters and could theoretically benefit from the efficient tensor CP structure, such as faster training and lower memory usage, these advantages are not yet realized due to the limitations of our current implementation. Future work will focus on optimizing this implementation. Future work will address these optimizations. However, the reduced parameter count already provides lower storage requirements and faster I/O.

We conducted hardware profiling to compare the performance of our LoRTA implementation against LoRA using HuggingFace PEFT. The results demonstrate negligible differences in resource consumption between the two methods. The slight gap in training time for LoRTA can be addressed through further optimizations, ranging from leveraging tools like Torch Compile, to implementing our CP tensor adapter model more efficiently.

| Rank | Method | GPU Mem. (GB) | FLOPs (avg) | MACs (avg) | Time (s/step) |
|------|--------|---------------|-------------|------------|---------------|
| 4    | LoRA   | 12.84         | 272         | 136        | 0.07          |
|      | LoRTA  | 12.88         | 272         | 136        | 0.14          |
| 64   | LoRA   | 13.08         | 276         | 138        | 0.09          |
|      | LoRTA  | 12.98         | 273         | 136        | 0.14          |

Table 11: Maximum GPU memory usage (GB), average FLOPs(GB), MACs(GB), and training time (seconds per step) for LoRA and LoRTA.

