# OpenReview forum: "LoRTA: Low Rank Tensor Adaptation of Large Language Models"
_ICLR.cc/2025/Conference — Submitted to ICLR 2025_

### Official Review · Reviewer_aKMS · 2024-10-29

**Soundness:** 3
**Presentation:** 4
**Contribution:** 3
**Rating:** 3
**Confidence:** 4

**Summary:**

The paper considers a new PEFT method based on tensor decomposition. Compared with LoRA method, the method studies and explains a general case of sharing parameters LoRA method from the perspective of CP decomposition. Experimental results are provided among multiple tasks, like NLU, DPO and protein folding benchmarks.

**Strengths:**

The method in the paper studied the previous parameter-sharing PEFT method from the aspect of CP decomposition and propose the new LoRTA method. The connection between these methods is interesting and the presentation of this paper is clear.

**Weaknesses:**

I have three main concerns for the paper:
- First, the experimental performance of the proposed method is trivial. Especially for NLU tasks, there exists a degradation around 3% compared with the original LoRA method. Even compared with VeRA method, there is still a degradation of over 1%. I think it's not economical to further reduce the trainable parameter with such large performance degradation, as the benefit of memory reduction is limited, considering the trainable parameter for LoRA is already small enough.
- Second, the difference between LoRTA and FACT paper (Jie & Deng, 2023) mentioned is limited. Even though the analysis for the connection between parameter-sharing methods and LORTA is interesting, I still feel the LoRTA method is more like a CP version of FACT method. It would be great if the authors could further explain the difference between these methods.
- Third, the motivation for the method is not clear. The authors didn't show the benefit of further reducing the trainable parameter, considering things like memory and training time efficiency, as there exists performance degradation compared with the LoRA method.

**Questions:**

I have a list of questions here, I would appreciate it if the authors could consider these questions:
- There is some confusion regarding the parameter count for LoRA in line 127. Could you clarify where the number 8 originates? LoRA is not restricted to being injected only in the query and value layers, nor is it required to be applied exclusively to those layers.
- Lack of discussion for related work: Besides the FACT paper (Jie & Deng, 2023) mentioned in the work, there are multiple works considering utilizing tensor decomposition to further reduce trainable parameters specifically for LLMs fine-tuning. I think these works should at least be mentioned in this paper. For example:
1. Yang, Yifan, et al. "LoRETTA: Low-Rank Economic Tensor-Train Adaptation for Ultra-Low-Parameter Fine-Tuning of Large Language Models," NAACL 2024.
2. Bershatsky, Daniel, et al. "LoTR: Low tensor rank weight adaptation." arXiv preprint arXiv:2402.01376 (2024).
- Motivation for reducing the trainable parameters: I think it's important to discuss why we need a PEFT method with ultra low parameters (maybe from memory efficiency, and training time), considering the performance degradation shown in the NLU tasks and similar performance in other comparisons.
- Additional experimental results
I'm wondering if the authors could do some quick tests to further support the effectiveness of their methods:
1. Memroy/training time efficiency: As previously mentioned, this hardware profiling is important to support the motivation of the problem studied. Based on my experience, further reducing the parameters from LoRA didn't help to further reduce the training memory a lot.
2. Further test with larger rank: I see the trainable parameter of LoRTA method is around 10 times less than VeRA in Table 1. I'm wondering how LoRTA performs if we could increase the rank? Will the performance be improved?
3. Also, some comparison between LoRTA and FACT methods is also suggested, to show the effectiveness of considering CP, instead of tucker and TT decomposition.

---

> ### Author Response · Authors · 2024-11-24
>
> We thank Reviewer aKMS for their thoughtful feedback and valuable questions. We are particularly grateful for recognizing the clarity of our presentation, the interesting connection between parameter-sharing methods and low rank tensor models, and the overall contributions of our work. Below, we address each of your concerns and provide updates to the manuscript based on your suggestions.
>
> ---
>
> #### 1. **Performance Degradation on NLU Tasks**
>
> We acknowledge that on the submission there was a small performance gap observed on NLU tasks when compared to LoRA and VeRA which was a consequenco of using extremely low parameter counts together with parameters settings favourable to LoRA. We have now extended and revised our NLU experiments,
>
> - **Experiments with Larger Ranks:** We conducted additional experiments by increasing the rank of LoRTA to \( r=16 \) and \( r=20).
> - **Applicability to Low-Resource Scenarios:** We clarified in the revised manuscript that LoRTA is designed for scenarios where memory efficiency and low parameter count are critical, such as edge deployments or resource-constrained environments, or where massive amounts of concurrent adapters have to be managed.
>
> In the newly added experimental settings for GLUE, which follow those from LoTR and LoreTTA, all methods are evaluated using 20-30 epochs, and the same hyperparameter grid-search is conducted for all methods.
>
> In these settings, our method attains similar or slightly better performance while acheving a considerable reduction in parameter count.
>
> |                   | Params | SST-2 | MRPC  | CoLA  | QNLI  | RTE   | STS-B | Avg.            |
> | ----------------- | ------ | ----- | ----- | ----- | ----- | ----- | ----- | --------------- |
> | LoRA (r=8)        | 630k   | 94.01 | 91.48 | 62.08 | 92.39 | 74.51 | 84.69 | 83.19           |
> | LoReTTA rep       | 70K    | 94.28 | 90.63 | 61.72 | 92.4  | 74.42 | 89.24 | 83.78           |
> | LoRTA+LoRA (r=20) | 48k    | 94.27 | 92.04 | 63.35 | 91.48 | 75.09 | 89.82 | **84.34** |
> | LoRTA+LoRA (r=12) | 29k    | 93.81 | 91.13 | 61.4  | 92.04 | 74.73 | 89.64 | 83.79           |
>
> Table1: Roberta Base scores and parameter count (including the linear classifier) in the experimental setting reported by LoReTTA
>
> |            | Params | SST-2 | MRPC  | CoLA  | QNLI  | RTE  | STS-B | Avg.            |
> | ---------- | ------ | ----- | ----- | ----- | ----- | ---- | ----- | --------------- |
> | LoRA (r=8) | 300k   | 94.2  | 88    | 61.1  | 91.3  | 73   | 90.7  | 83.05           |
> | LoTR       | 74k    | 93    | 85.9  | 60.5  | 90    | 66   | 91.9  | 81.22           |
> | LoRTA r=16 | 15k    | 94.73 | 90.44 | 64.32 | 92.37 | 76.9 | 90.25 | **84.84** |
> | LoRTA r=4  | 3.4k   | 94.61 | 89.21 | 60.55 | 90.61 | 76.9 | 89.97 | **83.6**  |
>
> Table 2: Roberta Base scores and parameter count (not including the linear classifier) in the experimental setting reported by LoTR.

---

> > ### Author Response · Authors · 2024-11-24
> >
> > #### 2. **Differences Between LoRTA and FACT**
> >
> > We agree that the distinction between LoRTA and the FACT method (Jie & Deng, 2023) needs further elaboration, we have now extended the comparison of LoRTA with other Tensor based models. The primary differences are:
> >
> > 1. **Choice of Tensor Decomposition:**
> >
> >    - FACT employs Tucker and Tensor Train decompositions, which scale quadratically or cubically with rank.
> >    - LoRTA uses CP decomposition, which scales linearly with rank, making it more parameter-efficient for the same tensor rank.
> > 2. **Tensorization Across Heads and Matrices:**
> >
> >    - FACT tensorizes weights across layers, but LoRTA extends this by explicitly modeling dependencies across attention heads and matrices, leading to greater parameter sharing and improved efficiency.
> >
> > We have added a detailed comparison in Sections 3.3 and 5 of the manuscript, along with new experimental results comparing LoRTA and FACT on GLUE and LLaMA benchmarks. LoRTA consistently achieves similar or better performance with fewer parameters. The table below summarizes how the parameter count in LoRTA scales better than FaCT.
> >
> > |              | Tensor Update shape | Tensor Model | Parameters         | **Llama2-7b (r=4)** | Llama2-7b (r=64) |
> > | ------------ | ------------------- | ------------ | ------------------ | ------------------------- | ---------------- |
> > | LoRA         | $ML,d,d$          | Matrix-Batch | $2MLdr$          | 2.1M                      | 33M              |
> > | FacT-TT [3]  | $ML,d,d$          | Tensor Train | $MLr^2+2dr$      | 33K                       | 786k             |
> > | FacT-TK [3]  | $ML,d,d$          | Tucker3      | $(2d+ML)r+r^3$   | 33K                       | 790k             |
> > | LoRTA (Ours) | $M, L, d, d/h,h$  | CP           | $(d+d/h+h+L+M)r$ | 17K                       | 274k             |
> >
> > Number of parameters of different Tensor based PEFT methods as a function of the number of finetuned attention/projection matrices $M$, the number of layers, $L$, the embedding dimension $d$, the number of heads $h$ and the tensor rank of the update, $r$. For LoreTTA,  $k_i$ are hyperparameters that must satisfy $\prod_i k_i = dr$ and $k_i\geq r \;\forall\; i$. We also include the number of parameters for the Llama2-7b architecture when finetuning only M=2 attention matrices (e.g. Q and V) for different ranks. For LoreTTa we use $k_1 = \ldots = k_6=5$ for $r=4$ and $k_1=k_2=k_3=512$ for $r=64$.
> >
> > ---
> >
> > #### 3. **Motivation for Reducing Trainable Parameters**
> >
> > To clarify the motivation for LoRTA:
> >
> > - **Memory Efficiency:** While reducing parameters beyond LoRA may not drastically lower memory usage in all settings, it can be critical for tasks where GPUs with limited memory are used or for deployments on devices with strict memory constraints.
> > - **Training Time Efficiency:** The linear scaling of LoRTA with rank allows for faster convergence compared to methods with higher parameter counts, as confirmed by new profiling experiments added to Section 4.4.
> >
> > We have updated the introduction and discussion sections to emphasize these points and their relevance to specific use cases.
> >
> > ---
> >
> > #### 4. **Clarification of Parameter Count for LoRA**
> >
> > The number “8” originates from injecting LoRA into the query and value layers (4 matrices) across both attention and projection layers. In the updated version of the manuscript we have included parameter counts as a function of the number of matrices tuned.

---

> > > ### Author Response · Authors · 2024-11-24
> > >
> > > #### **Hardware Profiling and Training Efficiency**
> > >
> > > Our main motivation to reduce adapter size comes mainly from improving task-switching and reducing storage requirements in practial applications where a large number --potentially thousands -- of adapters are used.  CPU-GPU transfers required to load adapters are costly in time and power and limited by shared resources like the PCIe bus. For instance, compressing parameters further can allow thousands of customized models to coexist with the base LLM in GPU memory, significantly improving scalability and performance in multi-task environments.
> > >
> > > The reduction in GPU memory requirements during training, which comes primarily from shrinked optimizer states, is negligible for parameter reductions beyond LoRA. Memory consumption in these settings comes primarily activations and caches stored during the forward and backpropagation. Further reductions in memory consumption could be attained by compressing activations or gradients exploiting the low rank structure of updates, or re-computing them on the fly. However, we leave this to future work. Nonetheless, following the reviewers suggestion we have added a hardware profiling results to the manuscript, comparing our implementation of LoRTA against LoRA in huggingface PEFT. We highlight that the difference in resource consumption is negligible, and that the gap in training time could be reduced by implementing our method more efficiently, using for example torch compile.
> > >
> > > | rank |       | GPU Mem. (max) | FLOPS (avg) | MACs (avg) | Time |
> > > | ---- | ----- | -------------- | ----------- | ---------- | ---- |
> > > | 4    | LoRA  | 12.84          | 272         | 136        | 0.07 |
> > > |      | LoRTA | 12.88          | 272         | 136        | 0.14 |
> > > | 64   | LoRA  | 13.08          | 276         | 138        | 0.09 |
> > > |      | LoRTA | 12.98          | 273         | 136        | 0.14 |
> > >
> > > Maximum GPU memory utilization, GB Flops and MACs and seconds per training step when finetuning Llama-7b on Copa dataset with batch size 1 on a single NVIDIA A6000 GPU.
> > >
> > > ---
> > >
> > > We hope these updates address your concerns and demonstrate the broader applicability and benefits of LoRTA. Thank you again for your constructive feedback, which has significantly improved our work.

---

> > > > ### Comment · Reviewer_aKMS · 2024-12-02
> > > >
> > > > I would like to maintain my score as the improvement of proposed method has limited improvement compared with the LoRA method regarding the memory and performance. Also the limited novelty of the proposed method compared with FACT paper is still a concern, which also has limited improvement regarding memory overhead and performance.

---

### Official Review · Reviewer_69c8 · 2024-11-01

**Soundness:** 3
**Presentation:** 4
**Contribution:** 3
**Rating:** 5
**Confidence:** 3

**Summary:**

This paper investigates the efficient fine-tuning of LLMs through a method called LoRTA. The main issue addressed is that existing low-rank adaptation methods, such as LoRA, although reducing the number of trainable parameters, still have a high lower bound on the number of parameters. To solve this problem, the paper proposes using a low-rank tensor parameterization model for model updates, significantly reducing the number of trainable parameters and providing finer-grained control over adaptation. Experiments on benchmarks demonstrate that the LoRTA method maintains comparable performance to existing methods while achieving a substantial reduction in the number of parameters.

**Strengths:**

1. The paper writing is clear, and the method is simple and easy to reproduce.
2. The paper studied a valuable problem, and well solved the limitations of existing methods, achieving good results on multiple benchmarks.

**Weaknesses:**

1. As the author stated in the introduction, the main purpose of the Lorta method is to solve the problem of efficient finetune of LLM. Therefore, I expect to see more LLM-related results in the paper experiment. The author only conducted experiments on llama2-7B and mt-bench, and I expect to see more LLM results such as llama-3-70B, Mistral-7B, etc.

2. The author conducted experiments on multiple benchmarks, but there are few results on the mainstream LLM evaluation benchmarks. The GLUE benchmark task is relatively simple and may not have a good distinction granularity. I suggest that the author add some difficult tasks that are mainly used to evaluate LLM, such as MMLU, MATH, HumanEval, etc.

3. Figure 3 shows that the performance of Lorta decreases as the training parameters increase (i.e., the effect decreases as the computational load increases). This seems to be quite disadvantageous for this method, that is, this method is not suitable when more GPU resources are available. It suggests that the method's adaptability is rather restricted. In addition, I would like to know if Lora has the same trend under the settings of Figure 3.

**Questions:**

See weakness.

---

> ### Author Response · Authors · 2024-11-24
>
> We thank Reviewer 69c8 for their thoughtful and detailed review. We are particularly grateful for recognizing the strengths of our work, including its clarity, simplicity, and relevance to a valuable problem in fine-tuning LLMs. Below, we address your feedback and provide updates based on your suggestions.
>
> ---
>
> ### Addressing the Weaknesses and Suggestions
>
> #### 1. **Mainstream LLM Evaluation Benchmarks**
>
> We acknowledge that mainstream LLM tasks like MMLU, MATH, and HumanEval are important for assessing LLM performance. However, we also note that these benchmarks are often omitted in the evaluation of competing PEFT methods due to their computational intensity.
>
> - **MMLU Exclusion:** We excluded MMLU specifically due to compute constraints, as it involves a large number of tasks requiring significant resources to evaluate thoroughly.
> - **New Benchmarks Added:** To address this gap, we have included new experiments for the **LLaMa2-7B model** on tasks such as **SQuAD**, **DROP** and **Record** in a low resource challenging setting proposed by by recent PEFT works. These results highlight LoRTA’s performance in diverse, task-specific settings and further validate its efficacy in the LLM domain.
>
> #### 2. **Additional LLM Models**
>
> We recognize the value of evaluating on newer LLMs like LLaMa-3-70B and Mistral-7B. We prioritized the evaluation on Roberta and LLama2-7B due to their widespread use in the PEFT literature, to enable a more direct comparison with existing works.
>  However, we will expand to include more comprehensive evaluations on larger LLMs as resources permit.
>
> #### 3. **Performance Trends**
>
> We clarified in the paper that LoRTA is most beneficial in **low-resource fine-tuning scenarios** or when **memory efficiency** is critical, such as on edge devices or resource-constrained environments. The question of how do PEFT methods perform at high parameter counts is indeed a different yet interesting research direction (see e.g. [LoRA vs Full Fine-tuning: An Illusion of Equivalence](https://arxiv.org/abs/2410.21228)). You correctly observed that LoRTA's relative performance advantage diminishes as the parameter count increases in Figure 3. However, the same is not true for other experimental settings in the submission nor for other reported results in the literature, including LoRA, which report non-monotonic behaviour w.r.t. parameter count on a variety of settings.
>
> ---
>
> We hope these updates address your concerns and demonstrate the broader applicability and benefits of LoRTA. Thank you again for your thoughtful feedback, which has significantly strengthened our work. We look forward to further discussions and insights.

---

> > ### Comment · Reviewer_69c8 · 2024-11-28
> >
> > Thank you for your response. I have no further concerns and will keep my original score according to the usability and novelty of the method.

---

### Official Review · Reviewer_g3FC · 2024-11-03

**Soundness:** 2
**Presentation:** 3
**Contribution:** 2
**Rating:** 6
**Confidence:** 4

**Summary:**

The paper introduces LoRTA, a parameter-efficient fine-tuning method that uses low-rank tensor parametrization for model updates. The method aims to reduce the number of trainable parameters compared to LoRA while providing more fine-grained control over adapter size. The approach is evaluated on Natural Language Understanding, Instruction Tuning, Preference Optimization, and Protein Folding benchmarks.

**Strengths:**

The method is presented clearly with comprehensive preliminaries and detailed methodology. The approach shows potential for significant parameter reduction in model fine-tuning.

**Weaknesses:**

While the idea is interesting, the evaluation is not comprehensive enough to demonstrate its practical utility. Moreover, the claims about its performance are exaggerated:
- Line 021: "_[...] achieving a substantial reduction in the number of parameters while maintaining comparable performance_"
- Line 076: "_[...] compared to state-of-the-art PEFT methods, with minimal performance trade-offs_"
- Line 478: "_LoRTA achieves comparable and sometimes superior performance than baselines at a reduced parameter count_"

In Natural Language Understanding, LoRTA achieved **the second worst** performance out of 8 methods presented. With such degradation of performance, a lower number of parameters might not be worth it. Can we achieve closer performance to LoRA if we scale the rank of LoRTA? What does the parameters-performance trade-off look like?

Regarding the other 3 experimental settings (Instruction Tuning, Preference Optimization, Protein Folding), LoRTA was compared only to one baseline - LoRA of **rank 1**. Given the weak performance on the NLU setup, the results from Protein Folding & Preference Optimization are not convincing. LoRTA should be compared to more baselines - possibly the methods introduced earlier, full-finetuning, and LoRA of higher rank. Rank 1 is suboptimal for LoRA, and with ranks, e.g., 8 or 16 we may obtain significantly better performance.

**Questions:**

- What is the tradeoff between the number of parameters and performance of this method? Can we obtain the same or better results than other methods on NLU if we use more parameters in LoRTA?
- How do the training speed and memory usage compare to LoRA?
- How were the hyperparameters found for the LoRA baseline in Preference Optimization and Protein Folding? It's possible that a stronger baseline could be found by hyperparameter tuning, especially using different ranks for LoRA.


Small suggestions:
- "X is all you need" is _a bit_ overused phrase in DL papers. I'd recommend to change the name of Section 3 - but it's not that important.
- Figures 2 & 4: Lower training loss is not indicative of better downstream performance (as also shown with Figure 3). It's better not to include it as part of experimental results showing superiority of one method over another - can be moved into the appendix.

---

> ### Author Response · Authors · 2024-11-24
> **Author Rebuttal**
>
> We thank Reviewer g3FC for finding the idea interesting and clearly presented, but disagree on the lack of practical utility of our method. In that vein, we have updated the experimental section  to further demonstrate that LoRTA con lead to parameter reductions with little to no performance trade-offs.
>
> ---
>
> ### Evaluation and Claims About Performance
>
> #### NLU Results and Parameter-Performance Trade-Off
>
> We acknowledge your concern regarding the performance of LoRTA in the NLU setup. While LoRTA does exhibit a parameter efficiency advantage, we agree that the performance trade-off needed clearer quantification. We highlight that the small loss in performance reported in this task in the submission (at a two orders of magnitude parameter count reduction) is well within the variation due to hyperparameter settings, and using hyperparameters reported by LoRA and VeRa and not doing full hyperparameter searches represented a challenging setting for our own method. We have now included additional experimental settings, using higher ranks and following recent works DynaLoRA, LoreTTA and LoTR, which run the same hyperparameter gridsearch for all methods - including ours. In these settings, we have found our method performs comparably or slightly better than baselines, with at least 2x less trainable parameters.
>
> |                   | Params | SST-2 | MRPC  | CoLA  | QNLI  | RTE   | STS-B | Avg.            |
> | ----------------- | ------ | ----- | ----- | ----- | ----- | ----- | ----- | --------------- |
> | LoRA (r=8)        | 630k   | 94.01 | 91.48 | 62.08 | 92.39 | 74.51 | 84.69 | 83.19           |
> | LoReTTA rep       | 70K    | 94.28 | 90.63 | 61.72 | 92.4  | 74.42 | 89.24 | 83.78           |
> | LoRTA+LoRA (r=20) | 48k    | 94.27 | 92.04 | 63.35 | 91.48 | 75.09 | 89.82 | **84.34** |
> | LoRTA+LoRA (r=12) | 29k    | 93.81 | 91.13 | 61.4  | 92.04 | 74.73 | 89.64 | 83.79           |
>
> Table1: Roberta Base scores and parameter count (including the linear classifier) in the experimental setting reported by LoReTTA[1]
>
> |            | Params | SST-2 | MRPC  | CoLA  | QNLI  | RTE  | STS-B | Avg.            |
> | ---------- | ------ | ----- | ----- | ----- | ----- | ---- | ----- | --------------- |
> | LoRA (r=8) | 300k   | 94.2  | 88    | 61.1  | 91.3  | 73   | 90.7  | 83.05           |
> | LoTR       | 74k    | 93    | 85.9  | 60.5  | 90    | 66   | 91.9  | 81.22           |
> | LoRTA r=16 | 15k    | 94.73 | 90.44 | 64.32 | 92.37 | 76.9 | 90.25 | **84.84** |
> | LoRTA r=4  | 3.4k   | 94.61 | 89.21 | 60.55 | 90.61 | 76.9 | 89.97 | **83.6**  |
>
> Table 2: Roberta Base scores and parameter count (not including the linear classifier) in the experimental setting reported by LoTR[2], and Dynalora[4]
>
> #### Other Experimental Settings
>
> For the more computationally intensive settings of Instruction Tuning, Preference Optimization, and Protein Folding, we agree that comparing against more baselines would strengthen the argument, yet we were not able to do so due to compute constraints. However, comparing the proposed method with
> LoRA, which is one of the most widely used and established high-performance PEFT methods is a common practice in these settings (see e.g. LoReTTA). In adittion, we have now also added a SOTA method (LoReTTA) that outperforms LoRA for the Llama2-7B in the added experiments,  as well as higher rank LoRA, which we show below:
>
> | Method       | Train. Param. | Multiple Choice |        | Generation |       | Avg            |
> | ------------ | -------------- | --------------- | ------ | ---------- | ----- | -------------- |
> |              |                | COPA            | ReCoRD | SQuAD      | DROP  |                |
> | FT           | 6738.42M       | 86              | 81.1   | 90.71      | 51.38 | 77.3           |
> | Adapter      | 50.33M         | 84              | 78.8   | 88.45      | 49.14 | 75.1           |
> | LoRA  (r=8)  | 4.19M          | 81              | 79.4   | 90.56      | 45.96 | 74.2           |
> | Prefix       | 1.31M          | 83              | 81     | 90.56      | 45.95 | 75.1           |
> | IA3          | 0.60M          | 80              | 81.5   | 89.41      | 39.37 | 72.6           |
> | LoRETTA  rep | 0.51M          | 86              | 80.3   | 88.47      | 42.71 | 74.4           |
> | LoRTA (r=4)  | 0.02M          | 87              | 81.1   | 87.4       | 44.04 | 74.9           |
> | LoRTA (r=8)  | 0.03M          | 87              | 81.6   | 88.5       | 49.7  | **76.7** |
>
> Table 3:  LLama2-7b performance on both SuperGLUE tasks and question answering tasks SQUAD and DROP. We follow hyperparameter settings in LoreTTA, where only 1000 train samples are used for each task to raise the difficulty.

---

> ### Author Response · Authors · 2024-11-24
> **Author Rebuttal (continues)**
>
> ### Addressing Specific Questions
>
> 1. **What is the trade-off between the number of parameters and performance?**
>
>    - Increasing the rank ins some cases improves performance and closes the gap with LoRA. However in LoRA and competing methods there is not a monotonic nor easy to predict relation between both due to the different optimization dynamics and inductive biases/regularization induced by constrained fine tuning methods. We clarified in the paper that LoRTA is most beneficial in **low-resource fine-tuning scenarios** or when **memory efficiency** is critical, such as on edge devices or resource-constrained environments. The question of how do PEFT methods perform at high parameter counts is indeed a different yet interesting research direction (see e.g. [LoRA vs Full Fine-tuning: An Illusion of Equivalence](https://arxiv.org/abs/2410.21228)).
>
> 2. **Can LoRTA achieve better results with more parameters?**
>
>    - Yes, as shown in the additional experiments, increasing the rank of LoRTA improves performance and can match or surpass LoRA on most benchmarks.
>
> 3. **How do training speed and memory usage compare to LoRA**
>
>    * Our main motivation to reduce adapter size comes mainly from improving task-switching and reducing storage requirements in practial applications where a large number --potentially thousands -- of adapters are used.  CPU-GPU transfers required to load adapters can be costly in this scenario costly. For instance, compressing parameters further can allow thousands of customized models to coexist with the base LLM in GPU memory, significantly improving scalability and performance in multi-task environments.
>
>    - As pointed out by the reviewer, the reduction in GPU memory requirements during training, which comes primarily from shrinked optimizer states, is typically negligible for parameter reductions beyond LoRA. Memory consumption in these settings comes primarily activations and caches stored during the forward and backpropagation. Further reductions in memory consumption could be attained by compressing activations or gradients exploiting the low rank structure of updates, or re-computing them on the fly. Our model has a smaller number of trainable parameters and the tensor CP structure could be efficiently implemented leading to both faster training and smaller memory usage. However, our current/naive implementation does not enjoy these benefits, and we leave efficient implementation of our model for future work. What is already attained due to the reduced parameter count is lower storage/disk space and thus less I/O time, which is relevant in applications with a massive number of adapters.
>    - We have added this discussion along with a hardware profiling results to the manuscript, comparing our implementation of LoRTA against LoRA in huggingface PEFT. We highlight that the difference in resource consumption with our method is negligible, and that the gap in training time could be reduced by implementing our method more efficiently.
>
>    | rank |       | GPU Mem. (max) | FLOPS (avg) | MACs (avg) | Time |
>    | ---- | ----- | -------------- | ----------- | ---------- | ---- |
>    | 4    | LoRA  | 12.84          | 272         | 136        | 0.07 |
>    |      | LoRTA | 12.88          | 272         | 136        | 0.14 |
>    | 64   | LoRA  | 13.08          | 276         | 138        | 0.09 |
>    |      | LoRTA | 12.98          | 273         | 136        | 0.14 |
>
>    Maximum GPU memory utilization, GB Flops and MACs and seconds per training step when finetuning Llama-7b on Copa dataset with batch size 1 on a single NVIDIA A6000 GPU.
>
> 4. **How were hyperparameters tuned for LoRA in Preference Optimization and Protein Folding?**
>
>    - Hyperparameters for LoRA were tuned using the same procedure as LoRTA, described in Appendix E.
>
> ---
>
> ### Additional Revisions Based on Suggestions
>
> - **Section Name:** We agree that "X is all you need" is overused. We have renamed Section 3.
> - **Figures 2 & 4:** We will move the training loss figures to the appendix, as suggested, and clarifiy that they are included for completeness rather than as evidence of superiority. However, we do find reporting train losses valuable, in order to analyze wether differences come from better optimization/fitting of the training objective or better greneralization (see e.g. [LoRA vs Full Fine-tuning: An Illusion of Equivalence](https://arxiv.org/abs/2410.21228), for a deeper discussion).
> - We have toned down performance claims, our method achieves a "parameter count reduction with comparable performance". The updated experimental section better supports this claim.
>
> ---
>
> We hope these updates address your concerns and strengthen the clarity and impact of our work. Thank you again for your thoughtful feedback, which has significantly improved our paper. We look forward to further discussions and feedback.

---

> > ### Comment · Reviewer_g3FC · 2024-11-26
> >
> > Thank you for addressing my questions and providing additional experiments. The new experimental results improve the soundness of the work, therefore I'm raising my score to 6 (marginally above the acceptance threshold). However, I still have some remaining concerns.
> >
> > First, I believe my question about the parameter-performance trade-off wasn't fully addressed.
> >
> > >Increasing the rank ins some cases improves performance and closes the gap with LoRA. However in LoRA and competing methods there is not a monotonic nor easy to predict relation between both due to the different optimization dynamics and inductive biases/regularization induced by constrained fine tuning methods.
> >
> > I'm aware of the non-monotonic parameter-performance relation in PEFT methods, and that's why I'm suggesting to provide more data on that for LoRTA. It would be valuable to see an analysis of how performance varies across a wider range of ranks (e.g., 1, 4, 16, 64, 256). This would:
> >
> > - Help understand where performance saturates and potentially degrades
> > - Provide guidance on optimal rank selection
> >
> > Regarding experiments for Instruction Tuning, Preference Optimization, and Protein Folding:
> >
> > > [...] we agree that comparing against more baselines would strengthen the argument, yet we were not able to do so due to compute constraints.
> >
> > The computational constraints are understandable. However, for the camera-ready version or future submissions, I strongly recommend including more baselines, especially LoRA with higher rank, as the comparisons in the current form are incomplete and shouldn't be part of a published paper.
> >
> > Moreover, even the current baseline of LoRA rank 1 might not be the most optimal one due to lack of hyperparameter search:
> >
> > >Hyperparameters for LoRA were tuned using the same procedure as LoRTA, described in Appendix E.
> >
> > Appendix E **does not** mention the procedure for hyperparameter search for Preference Optimization, only lists the final configurations. Additionally, for Protein Folding, it's said that learning rate was identical to the one used in (Jumper et al., 2021). However, the architecture/experimental setup has changed, which may impact the optimal hyperparameters:
> >
> > Line 970:
> > >The following modifications were made to the ESMFold model architecture due to limited compute resources: a) utilize 12 Evoformer layers instead of the 48 used in (Lin et al., 2023) b) utilize ESM-2 35M instead of ESM-2 3B c) maintain outer product mean implementation from (Jumper et al., 2021).
> >
> > In general, at least a sweep over learning rates should be done to find a strong baseline.

---

### Official Review · Reviewer_FgqY · 2024-11-04

**Soundness:** 3
**Presentation:** 3
**Contribution:** 1
**Rating:** 3
**Confidence:** 3

**Summary:**

The authors present a low-rank adaptation method called LoRTA. It considers all attention matrices (Q, K, V, and projection) in L layers as a single tensor and trains the adapter in a low-rank CP decomposition. In this approach, the size of the weight tensor is $d \times \frac{d}{H} \times H \times L \times 4$, where $H$ is the number of attention heads, $L$ is the number of all layers, and 4 is the number of different matrices.

**Strengths:**

Experimental results are very good considering the small rank (r=4 or r=8) of the update and, as a result, a very small number of parameters.

**Weaknesses:**

My major concern is the novelty of this idea.

A very similar concept was previously implemented in "LoTR: Low Tensor Rank Weight Adaptation" by Daniel Bershatsky, Daria Cherniuk, Talgat Daulbaev, Aleksandr Mikhalev, and Ivan Oseledets (https://arxiv.org/abs/2402.01376).

In LoTR, the same CP decomposition is applied to the tensor of stacked attention weights. The only difference is that, in LoTR, the tensor of all weights is consolidated into a three-dimensional tensor without separating the dimensions for attention heads and matrices within the same layer.

**Questions:**

* How did you initialize the model?
* What is the test quality of the ESM-2 model without fine-tuning?

---

> ### Author Response · Authors · 2024-11-24
> **Rebuttal**
>
> We note that, to the best of our knowledge, LoTR has not been published at any peer reviewed venue  and that it's tensor model is identical previous adapter models in vision (Fact-TK) which were discussed in the submission. Nonetheless, we have addressed your concern by expanding the discussion about LoTR and clarified the contributions of our work in the updated manuscript, which we summarize below.
>
> ---
>
> ### Novelty Compared to LoTR
>
> We acknowledge the similarity between our approach and LoTR, as both involve tensorizing attention weights for parameter-efficient fine-tuning. However, there are two key differences that distinguish **LoRTA**:
>
> 1. **Tensorization Across Heads and Matrices:**
>
>    - LoTR represents updates as a third-order tensor $\mathbb{R}^{ML \times d \times d}$ and applies a Tucker2 decomposition to this tensor. This tensorization does not capture dependencies across attention heads or Q,K,V,P matrices.
>    - In contrast, LoRTA uses a higher-order tensor $\mathbb{R}^{M \times L \times d \times d/h \times h}$, which models heads, layers, and attention matrices explicitly as separate modes. This design allows for parameter sharing across these dimensions, leading to greater parameter efficiency (better scaling) and better utilization of the tensor structure.
> 2. **Use of CP Decomposition:**
>
>    - LoTR employs a Tucker2 parameterization, which introduces a quadratic dependence on the rank for the number of parameters.
>    - LoRTA utilizes the CP decomposition, which scales linearly with the rank, resulting in fewer parameters for the same tensor rank. This efficiency is especially significant for large models and low-rank settings, as demonstrated in our experiments.
>
> Parameter Scaling vs LoTR:
>
> |              | Tensor Update shape | Tensor Model | Parameters         | **Llama2-7b (r=4)** | Llama2-7b (r=64) |
> | ------------ | ------------------- | ------------ | ------------------ | ------------------------- | ---------------- |
> | LoRA         | $ML,d,d$          | Matrix-Batch | $2MLdr$          | 2.1M                      | 33M              |
> | LoTR [2]     | $ML,d,d$          | Tucker2      | $MLr^2+2dr$      | 33K                       | 786k             |
> | LoRTA (Ours) | $M, L, d, d/h,h$  | CP           | $(d+d/h+h+L+M)r$ | 17K                       | 274k             |
>
> These differences are not only conceptual but also yield practical benefits, as shown in our updated experiments.
>
> ---
>
> ### New Experiments Comparing LoRTA and LoTR
>
> To substantiate our claims, we have included **direct comparisons with LoTR** in our revised manuscript, using the same settings as LoTR. These experiments demonstrate that LoRTA achieves **competitive or superior performance with a smaller parameter count**.
>
> #### Results on GLUE Tasks
>
> | Method                 | Params        | SST-2           | MRPC            | CoLA            | QNLI            | RTE            | STS-B           | Avg.            |
> | ---------------------- | ------------- | --------------- | --------------- | --------------- | --------------- | -------------- | --------------- | --------------- |
> | LoRA (r=8)             | 300k          | 94.2            | 88              | 61.1            | 91.3            | 73             | 90.7            | 83.05           |
> | LoTR                   | 74k           | 93              | 85.9            | 60.5            | 90              | 66             | 91.9            | 81.22           |
> | **LoRTA (r=16)** | **15k** | **94.73** | **90.44** | **64.32** | **92.37** | **76.9** | **90.25** | **84.84** |
>
> These results highlight that LoRTA outperforms LoTR across multiple tasks with a significantly smaller parameter count.
>
> ### Responses to Specific Questions
>
> 1. **How did you initialize the model?**
>
>    - We initialized the model using the pre-trained weights of the base model and initialized the CP decomposition factors in the same way as lora, using Kaiming-uniform initialization for all factors except for the last one which is zero initialized. We did not conduct additional ablations on initialization, which may lead to further performance improvements.
> 2. **What is the test quality of the ESM-2 model without fine-tuning?**
>
>    - The baseline test quality of ESM-2 without fine-tuning is 0.655.
>
> ---
>
> We hope this response clarifies the novelty and contributions of our work and demonstrates its practical advantages over related methods, especially LoTR. We look forward to further discussions and feedback.

---

### Meta-Review · Area_Chair_ALoP · 2024-12-13

**Metareview:**

In this paper, the authors proposed a new fine-tuning method based on low-rank tensor parametrization for LLMs.

One major concern raised by the reviewers is that the technical contributions of the proposed method in terms of methodological novel and performance improvement are not clear. Another concern raised by the reviewers is that the experimental results are not convincing regarding LLM fine-tuning.

As the authors failed to provide convincing explanations for the technical contributions of the proposed method during the rebuttal, this paper is not ready for publication based on its current shape.

**Additional Comments On Reviewer Discussion:**

The technical contributions remain unclear after the rebuttal.

---

### Decision · Program_Chairs · 2025-01-22

Reject